# A Comparative Study on the Effects of Different Sources of Carboxymethyl Poria Polysaccharides on the Repair of DSS-Induced Colitis in Mice

**DOI:** 10.3390/ijms24109034

**Published:** 2023-05-20

**Authors:** Zhijie Tan, Qiaoyi Zhang, Rou Zhao, Ting Huang, Yun Tian, Yuanshan Lin

**Affiliations:** 1College of Bioscience and Biotechnology, Hunan Agricultural University, Changsha 410128, China; 2Agricultural Bioengineering Institute, Changsha 410128, China

**Keywords:** carboxymethyl, *Poria cocos*, polysaccharide, colitis, intestinal flora

## Abstract

Carboxymethyl poria polysaccharide plays important anti-tumor, antioxidant, and anti-inflammatory roles. Therefore, this study aimed to compare the healing impacts of two different sources of carboxymethyl poria polysaccharides [Carboxymethylat Poria Polysaccharides I (CMP I) and Carboxymethylat Poria Polysaccharides II (CMP II)] on ulcerative colitis in mice caused by dextran sulfate sodium (DSS). All the mice were arbitrarily split into five groups (*n* = 6): (a) control (CTRL), (b) DSS, (c) SAZ (sulfasalazine), (d) CMP I, and (e) CMP II. The experiment lasted for 21 days, and the body weight and final colon length were monitored. A histological analysis of the mouse colon tissue was carried out using H&E staining to assess the degree of inflammatory infiltration. The levels of inflammatory cytokines [interleukin-1β (IL-1β), interleukin-6 (IL-6), tumor necrosis factor-α (TNF-α), and interleukin-4 (IL-4)] and enzymes [superoxide dismutase (SOD) and myeloperoxidase (MPO)] in the serum were examined using ELISA. Additionally, 16S ribosomal RNA sequencing was used to analyze the microorganisms in the colon. The results indicated that both CMP I and CMP II alleviated weight loss, colonic shortening, and inflammatory factor infestation in colonic tissues caused by DSS (*p* < 0.05). Furthermore, the ELISA results revealed that both CMP I and CMP II reduced the expression of IL-1β, IL-6, TNF-α, and MPO, and elevated the expression of IL-4 and SOD in the sera of the mice (*p* < 0.05). Moreover, 16S rRNA sequencing showed that CMP I and CMP II increased the plenitude of microorganisms in the mouse colon relative to that in the DSS group. The results also indicated that the therapeutic effect of CMP I on DSS-induced colitis in the mice was superior to that of CMP II. This study demonstrated that carboxymethyl poria polysaccharide from *Poria cocos* had therapeutic effects on DSS-induced colitis in mice, with CMP I being more effective than CMP II.

## 1. Introduction

Inflammatory bowel disease (IBD) is a long-term, recurrent inflammatory disorder of the gastrointestinal tract, typically classified as ulcerative colitis (UC) or Crohn’s disease (CD) [1]. As per the WHO’s data, colorectal cancer, with 1.8 million new cases worldwide, resulted in 880,792 deaths in 2018 [2]. In China, colorectal cancer became the second-most-common type of cancer in 2022, with a significant increase in its incidence [3]. A combination of genetics and the environment are thought to contribute to IBD, leading to the breakdown of the intestinal epithelial barrier and a disproportion of the intestinal flora [4]. Abdominal discomfort, loose stools and weight loss are the primary clinical manifestations of IBD [5]. UC is characterized by confluent inflammation of the colonic mucosa, which can involve any part of the gastrointestinal tract, from the mouth to the anus [6]. In recent decades, notable advancements have been made in basic and clinical research on the pathogenesis of IBD. However, due to its complex pathogenesis and the large individual differences, the treatment of UC still presents great challenges.

The colon is inhabited by approximately 3 × 10^13^ bacteria, which have a continuous interaction with the gut epithelium and are vital for health and disease [7,8]. The gut microbiome can alter host physiology by producing metabolites, such as secondary bile acids, folic acid, indole, trimethylamine-n-oxide (TMAO), neurotransmitters (e.g., serotonin), and short-chain fatty acids (SCFAs) [9,10]. Studies have shown that imbalances in the intestinal flora are also connected to the development and progression of chronic metabolic diseases (CMDs) [11,12], and that the SCFAs/gut microbiome associated with hypertension may be linked to sleep apnea and other disorders [13]. The vital microbiota secretes many kinds of metabolites and bacteriocins, which can influence the onset and progression of diseases [14].

The majority of drugs for UC can relieve clinical symptoms to some extent, but these clinical symptoms are prone to recurrence and have certain side effects that interfere with metabolism and immune response, often causing serious adverse effects [15,16]. Therefore, it is necessary to find new therapeutic drugs with higher safety and fewer side effects. In recent years, the pharmacological mechanisms and targets of action of a series of potential natural polysaccharides—including plant, animal, and fungal polysaccharides—have been reported [17], among which fungal polysaccharides have shown excellent anti-inflammatory [18,19], antioxidant [20,21], and anti-tumor effects [22], providing guidance for the development of novel clinical anti-inflammatory drugs. *Poria cocos*, belonging to the family Poriferae, is considered one of the most versatile fungi, with its most important active ingredient being Poria polysaccharide. Poria polysaccharides are composed of ribose, arabinose, xylose, mannose, glucose, and galactose [23]. It has been reported that, after various chemical modifications, Poria polysaccharides have higher solubility in water and greater anticancer effects [24,25]. As the cultivation of natural *Poria cocos* consumes a large amount of wood and causes great damage to the environment, the search for new production methods of *Poria cocos* has become a research hotspot, and liquid fermentation is expected to be an alternative to traditional cultivation methods. In this research, we studied the influence of two different sources of Poria polysaccharides [(carboxymethylat Poria polysaccharides I (CMP I), extracted by the liquid fermentation of *Poria cocos*)], and [(carboxymethylat Poria polysaccharides II (CMP II), extracted by natural *Poria cocos* nuclei)] on the repair of dextran sulfate sodium (DSS)-induced acute colitis in mice and on the intestinal microbiota of mice.

## 2. Results

### 2.1. Pathological Observation of the Colon

As shown in Figure 1, the histological examination demonstrated that the colon tissue of the DSS-treated mice was damaged by inflammatory cell infiltration, consisting mainly of neutrophils, lymphocytes, and macrophages (Figure 1B). No inflammation was noted in the CTRL group (Figure 1A), relative to the DSS group; the mice treated with SAZ, CMP I, and CMP II (Figure 1C–E) showed inflammation in the colon tissue, but the intensity decreased significantly. It was clear that the effect of CMP I (Figure 1D) was significantly greater than that of CMP II (Figure 1E).

### 2.2. Inhibitory Effects of Different Sources of Poria Cocos Polysaccharides on DSS-Induced Weight Loss and Colonic Length Reduction in Mice

The mice were given 5% DSS treatment for 7 days, followed by 7 days of treatment with various sources of Poria polysaccharide. The changes in their body weight and final colon length were measured (Figure 2A,B). As shown in Figure 2A, SAZ, CMP I, and CMP II, the treatments significantly improved DSS-induced body weight loss in mice, with the weight loss rate significantly decreasing on day 3, following SAZ treatment (*p* < 0.05). Figure 2B shows that compared to the DSS group, the SAZ, CMP I, and CMP II treatments significantly restrained the reduction in the colon length caused by DSS (*p* < 0.05), and the effect of CMP I was greater than that of CMP II (*p* < 0.05).

### 2.3. The Influence of the Expression Quantity of Inflammatory Cytokines and Two Enzymes in Mice Serum

Tissue injury is a crucial manifestation of colitis and involves excessive oxidative stress and an aberrant expression of the inflammatory factor. In comparison to the CTRL group, the DSS group experienced a significant decline in serum superoxide dismutase (SOD) and interleukin-4 (IL-4) expression (*p* < 0.05) and a significant rise in interleukin-1 1β (IL-1β), interleukin-6 (IL-6), tumor necrosis factor-α (TNF-α), and myeloperoxidase (MPO) expression (*p* < 0.05). The treatment with SAZ, CMP I, and CMP II all significantly reversed the imbalance of SOD, MOP, IL-1, βIL-6, TNF-α, and IL-4 expression (*p* < 0.05), with Group CMP I being more productive than group CMP II for this reversal (*p* < 0.05) (Figure 3A–F).

### 2.4. The Effect on Colonic Microbial Alpha Diversity in Mice

The V3-V4 region of 16S rRNA was analyzed through sequencing from collected samples of colon contents; Figure 4 shows the analysis of α-diversity in each group (Figure 4A–D). The analysis demonstrated that the abundance and diversity of colonic microbiota in the DSS group decreased significantly compared to the CTRL group (*p* < 0.05). The data indicated that compared to the DSS group, the Ace index and Chao1 index of the SAZ group increased significantly (*p* < 0.05), while the Simpson index and Shannon index did not change significantly. Although the four indexes of the CPMP I group were not significant, they exhibited an increasing trend; the four indices of the CMP II group did not differ much, and they did not exhibit a significant increasing trend. Overall, the restoration of intestinal microbial diversity in the mice with CMP I was greater than that with CMP II.

### 2.5. Effects on Different Levels of Microbial Abundance

#### 2.5.1. Effect on Microbial Abundance at the Phylum Level

Figure 5A,B show the top ten microorganisms at the phylum level, with the dominant phyla being Firmicutes, Bacteroidota, Verrucomicrobiota, and Proteobacteria. The proportions of Firmicutes in the CTRL, DSS, SAZ, CMP I, and CMP II groups were 42.68, 51.39, 41.07, 36.51, and 52.51%, respectively; Bacteroidota had values of 41.01, 24.61, 39.43, 44.74, and 34.40%; Verrucomicrobiota achieved 2.1, 4.67, 6.37, 6.74, and 1.94%; and for Proteobacteria the levels were 2.69, 9.58, 2.39, 3.62, and 1.67%. In Figure 5C, the relative number of Firmicutes in the CMP I group was significantly lower compared to the DSS group (*p* < 0.05), and the decrease in Firmicutes abundance in the other groups, except for in the CMP II group, was not significant (*p* > 0.05), but a decreasing trend could still be seen. In Figure 5D, the relative abundance of Bacteroidota is significantly elevated (*p* < 0.05) in the CTRL, SAZ, and CMP I groups relative to the DSS group, and although it was not significant in the CMP II group, it still increased to some extent. Interestingly, the content of Bacteroidota was greater in the CMP I group than in the CMP II group (*p* < 0.05). In Figure 5E, relative to the DSS group, SAZ, CMP I, and CMP II could significantly reduce Proteobacteria in the colon of mice (*p* < 0.05).

#### 2.5.2. Effect on Microbial Abundance at the Class Level

Figure 6A,B show the microorganisms that account for the top ten at the class level. The top three microorganisms in each group and their proportions were CTRL (Clostridia, 39.68%; Bacteroidia, 41.01%; Campylobacter, 6.6%), DSS (Clostridia, 43.82%; Bacteroidia, 24.61%; Bacilli, 7.57%), SAZ (Clostridia, 34.20%; Bacteroidia, 39.43%; Bacilli, 6.86%), CMP I (Clostridia, 29.33%; Bacteroidia, 44.75%; Bacilli, 7.16%), and CMP II (Clostridia, 48.83%; Bacteroidia, 32.40%; Desulfovibrionia, 4.11%). The results in Figure 6C show that the abundance of Clostridia in the CTRL, SAZ, and CMP I groups compared to the DSS group was not significantly different (*p* > 0.05), but the content was somewhat lower. Interestingly, the number of Clostridia increased in the CMP II group, instead, with a significant difference in content from the CMP I group (*p* < 0.05). The results in Figure 6D show that the CTRL, SAZ, and CMP I groups all significantly increased the number of Bacteroidia in the mouse colon relative to the DSS group (*p* < 0.05), while the increasing effect of CMP II was not significant (*p* > 0.05); nevertheless, the number still tended to increase. The results in Figure 6E show that the SAZ and CMP I groups could increase the abundance of Verrucomicrobiae in the mouse colon to some extent, relative to the DSS group (*p* > 0.05), while the abundance of Verrucomicrobiae in the CMP II group had a tendency to decrease.

#### 2.5.3. Effect on Microbial Abundance at the Order Level

Figure 7A,B show the microorganisms that accounted for the top ten in the order level. The relative abundance of the top three microorganisms in the CTRL, DSS, SAZ, CMP I and CMP II groups were Bacteroidales (40.95, 20.60, 39.31, 44.74, and 34.40%), Lachnospirales (25.44, 27.70, 24.87, 19.45, and 35.05%), and Oscillospirales (10.87, 9.91, 7.05, 8.60, and 11.76%). The results of Figure 7C showed that compared with the DSS group, the SAZ and CMP I groups could significantly improve the number of Bacteroidales in the mouse colon (*p* < 0.05), while the increase in the abundance of Bacteroidales in the CMP II group was not significant (*p* > 0.05), but still showed an increasing trend. The DSS, SAZ and CMP I groups could reduce the number of Lachnospirales in the mouse colon to a certain extent (*p* > 0.05). Interestingly, as shown in Figure 7D, the abundance of Lachnospirales in the mouse colon in the CMP II group did not decrease, but instead tended to increase. Additionally, there were significant differences between the two groups (*p* < 0.05). Figure 7E shows that the SAZ and CMP I groups exhibited increases in the abundance of Verrucomicrobiae in the mouse colon to some extent, compared with the DSS group (*p* > 0.05), while the abundance of Verrucomicrobiae in the CMP II group had a tendency to decrease.

#### 2.5.4. Effect on Microbial Abundance at Family Level

Figure 8A,B show the top ten microorganisms in terms of percentage at the family level. In the CTRL group, Lachnospiraceae (25.44%), Muribaculaceae (20.73%), and Helicobacteraceae (6.62%) were the most abundant microorganisms. Lachnospiraceae (27.70%), (MISSING) Bacteroidaceae (12.40%) and Muribaculaceae (7.34%) were the most abundant microorganisms in the DSS group. The most abundant microorganisms and their relative abundance in the SAZ, CMP I, and CMP II groups were Lachnospiraceae (24.87, 19.45, and 35.04%), Bacteroidaceae (20.40, 20.49, and 15.94%) and Muribaculaceae (13.43, 18.58, and 10.13%), respectively. According to the results in Figure 8C compared to the DSS group, SAZ and CMP I could reduce the abundance of Lachnospiraceae to a certain extent, while CMP II slightly increased the abundance of Lachnospiraceae, but none of them were remarkable (*p* > 0.05). Figure 8D indicated that relative to the DSS group, SAZ, CMP I, and CMP II all augmented the relative number of Muribaculaceae in the colon of mice, and the effects of SAZ and CMP I were significant (*p* < 0.05), while the effects of CMP II were not significant (*p* > 0.05). Interestingly, the number of Muribaculaceae in the CMP I group was higher than that in the CMP II group, with a significant difference (*p* < 0.05). Figure 8E showed that, compared to the DSS group, SAZ, CMP I, and CMP II all improved the relative number of Bacteroidaceae in the colon of mice, and the effects of SAZ and CMP I were significant (*p* < 0.05), while the effects of CMP II were not significant (*p* > 0.05).

#### 2.5.5. Effect on Microbial Abundance at Genus Level

Figure 9A,B show the top ten microorganisms at the genus level. In the CTRL group, *Muribaculaceae* (17.91%), *Lachnospiraceae* (11.48%), and the *Lachnospiraceae_NK4A136_group* (11.09%) were the most numerous microorganisms. In the DSS group, *Lachnospiraceae* (13.34%), *Bacteroides* (12.40%) and the *Lachnospiraceae_NK4A136_group* (8.44%) were the most abundant microorganisms. The top three microorganisms in SAZ, CMP I, and CMP II were *Lachnospiraceae* (23.60, 17.53, and 13.58%), *Bacteroides* (12.43, 20.49, and 15.94%), and *Muribaculaceae* (12.43, 23.60 and 8.57%), respectively. As shown in Figure 9C, compared with the DSS group, SAZ, CMP I and CMP II could all improve the number of *Bacteroides* in the mouse colon, and the effect of SAZ and CMP I was significant, while the effect of CMP II was not significant (*p* > 0.05). Figure 9D showed that relative to the DSS group, SAZ, CMP I and CMP II could increase the number of *Muribaculaceae* in the mouse colon, and the effect of CMP I was significant (*p* < 0.05), while the impact of SAZ and CMP II was not significant (*p* > 0.05). There was also a significant difference in the abundance of *Muribaculaceae* between the CMP I and CMP II groups (*p* < 0.05), and the content of *Muribaculaceae* in the CMP I group was superior to that in the CMP II group. Figure 9E showed that compared with the DSS group, SAZ, CMP I, and CMP II could improve the abundance of colon microbial *Parasutterella* in mice, and the influence of SAZ and CMP II was significant (*p* < 0.05), while the effect of CMP I was not significant (*p* > 0.05). The amount of *Parasutterella* in the CMP I and CMP II groups was significantly different (*p* < 0.05), and the content in the CMP I group was lower than that in the CMP II group.

### 2.6. Beta-Diversity and LEfSe Analysis of the Intestinal Microbial Flora

The V3-V4 region of 16S rRNA was analyzed through sequencing collected samples of colon contents, and the V3-V4 region of 16s rRNA was analyzed through sequencing collected samples of colon contents. Figure 10 shows the β diversity and LEfSe analysis of the colon microflora of mice. Figure 10A is the Venn diagram obtained through UTO clustering. It can be seen from the figure that the specific OUT numbers of CTRL, DSS, SAZ, CMP I, and CMP II were 2129, 1061, 1477, 1397, and 925, respectively. The coinciding OUT number of the five groups was 238. This indicated that there were differences in the species and number of microorganisms between the five groups. Our analysis of β diversity included Principal Co-ordinates Analysis (PCoA), Non-metric multidimensional scaling (NMDS), and Partial Least-Squares Discriminant Analysis (PLS-DA) to study the similarity of the overall microbial community structure (Figure 10B–D). In the analysis of PCoA, the top two major components accounted for 53.47% and 14.89%, respectively. The results of the PCoA analysis showed that the DSS treatment dramatically changed the community structure in the colon of mice, and the community structure after the SAZ, CMP I and CMP II treatment was significantly different from that of DSS, suggesting that all three treatments could improve the imbalance in the microbial community structure induced by DSS. Similar results were observed for NMDS (Figure 10C) and S-DA (Figure 10D). An LEfSe analysis was performed to understand the key microbial species in the different groups, and the results are shown in Figure 10E,F. The main marker bacteria in the CTRL group were *Muribaculaceae*, Prevotellaceae, *Lachnospiraceae*, and *Acteroidales*. The main key bacteria in the DSS group were Proteobacteria, *Parasutterella*, and Burkholderiales. The key bacteria in the SAZ group were *Bacteroides*, and the key bacteria in the CMP I group were *Bacteroidales* and Erysipelotrichales. The key bacteria in the CMP II group were *Lachnospiraceae*.

### 2.7. Correlation Analysis between Intestinal Microbe Difference and Serum Inflammatory Factors and Enzymes

To explore the relationship between gut microbes and serum enzymes and various inflammatory factors in mouse colitis, we investigated the correlation between microbes in the mouse colon and various inflammatory factors and enzymes in serum (Figure 11A–P). As shown in Figure 11, Bacteroidota was negatively correlated with TNF-α and IL-1β, but positively correlated with SOD and IL-4. Proteobacteria were positively correlated with IL-1β, IL-6, TNF-α, and MPO, but negatively correlated with IL-4 and SOD. *Parasutterella* was positively correlated with IL-1β, IL-6, and TNF-α, but negatively correlated with IL-4 and SOD. Bacteroidia was positively correlated with IL-4. These results indicated that Bacteroidota and Bacteroidia could promote the repair of colitis in mice, while Proteobacteria and *Parasutterella* increased the level of inflammation. According to the above results, CMP I can significantly improve the abundance of Bacteroidota and Bacteroidia in the colon of mice (*p* < 0.05). Moreover, Proteobacteria were significantly inhibited (*p* < 0.05) and *Parasutterella* abundance was inhibited to a certain extent (*p* > 0.05). CMP II could improve the abundance of Bacteroidota and Bacteroidia to a certain extent (*p* > 0.05), and significantly inhibit the abundance of Proteobacteria and *Parasutterella* (*p* < 0.05).

## 3. Discussion

The intestine is the largest digestive organ in the body [26]. IECs not only form a physical barrier between the host and its symbionts, but also enable their interactions [27]. Mucosal barrier defects, including abnormal cup cells, inflammatory gene expression, and increased numbers of IFNγ-expressing intraepithelial lymphocytes [28] can compromise the integrity of the intestinal mucosal barrier, which is necessary for the absorption of nutrients, for resistance to the invasion of intestinal pathogens, and for the maintenance of mucosal immunity [29,30]. To maintain the integrity of the intestine, epithelial cells need to be constantly renewed [31]. Increasing evidence suggests that failure to dysbiosis of proper intestinal flora can result in impaired function and increased permeability of the intestinal immune system, which can lead to further intestinal diseases [32,33,34]. This study’s findings indicated that the disruption of colonic epithelial cells and mucosal barrier in DSS-treated mice caused severe weight loss and colonic shortening. *Poria cocos* polysaccharides CMPI and CMPII from different sources had therapeutic and restorative effects on the DSS-induced disruption of colonic epithelial cells and mucosal barrier in mice, and it significantly inhibited the DSS-induced body weight reduction and colonic shortening in mice, thus reducing the inflammatory symptoms in mice. However, the effect of CMPI was greater than that of CMPII.

Inflammatory factors are important indicators to judge the severity of inflammation. Studies have shown that the expression of pro-inflammatory cytokines (such as IL-1β, IL-6, TNF-α, etc.) is significantly increased during intestinal inflammation, leading to a significant increase in intestinal epithelial tight-junction barrier permeability as a result of multiple effects. On the one hand, pathway activator NEMO or upstream NF-kB-activating kinases IKK1 and IKK2 lead to IEC-specific ablation to increase intestinal barrier permeability [35,36]; on the other hand, the activation of substrate-activated transcription factor ATF-2 by p38 kinase can further activate MLCK to increase intestinal inflammation [37]. In addition, intestinal barrier permeability can be increased by the activation of tight-junction barrier transmembrane proteins of the intestinal epithelium, such as the occludin, which increases the occlusal level of the Caco-2 monolayer and the intestinal epithelial cells of mice [38]. Anti-inflammatory cytokines (such as IL-4, IL-10, IL-22, etc.) can inhibit the proliferation of TH17 and TH2 cells [39] and intestinal wall disorders, thus alleviating colitis in mice. LPS is a major bacterial virulence factor [40,41]. LPS triggers a strong pro-inflammatory response and stimulates the exudation of pro-inflammatory factors (IL-1β, TNF-α, etc.) by host cells, further altering intestinal permeability [42]. However, the anti-inflammatory factors restored intestinal permeability by decreasing the expression quantity of pro-inflammatory cytokines [43]. Myeloperoxidase (MPO) is a heme-containing peroxidase that is mainly present in neutrophils and is involved in various inflammatory diseases [44]. Elevated MPO levels have been linked to increased inflammation and oxidative stress. Superoxide dismutase (SOD), an endogenous antioxidant enzyme, plays an important role in reducing oxidative stress [45]. Our results showed that both CMP I and CMP II significantly decreased the expression quantity of pro-inflammatory factors (IL-1β, IL-6, TNF-α) and myeloperoxidase (MPO) (*p* < 0.05), and elevated the expressions of anti-inflammatory factors IL-4 and superoxide dismutase (SOD) (*p* < 0.05) in the sera of mice induced by DSS. The repair impact of CMP I on DSS-caused colitis in mice was greater than that of CMP II.

Microbes in the human gut are key factors in the host metabolism, and are considered a potential source of novel therapies [46]. Research estimates indicate that there are as many bacteria in the gastrointestinal tract as there are cells that make up the human body [47]. The digestive tract has two different microbial ecosystems: the intracavitary microbiome, and the mucosal microbiome [48]. In the lumen, the main phyla are the Firmicutes and the Bacteroidetes, while the secondary phyla are the Actinobacteria, the Proteobacteria, and the Verrucomicrobia. In contrast, the number and diversity of bacteria in the mucosal layer are low, and the composition is clearly different [49]. The intestine is characterized by a specific microbiota: the enterotype, which is the permanent intestinal microbial system. Interestingly, the most abundant microbiota of the digestive tract is located on the mucosal surface of the colon [50]. The gut microbiota can change the severity of intestinal inflammation by altering microorganisms or their effectors (e.g., lipids, small molecules, or sugars) [51]. Changes in the interaction between intestinal epithelial cells and microbiome are important steps in the pathogenesis of IBD [52]. Increased levels of gamma-proteobacteria will induce inflammation [53]. The gut is a stable and methodical symbiotic environment; unfavorable conditions can disrupt this stability [54,55]. Our study revealed that the colonic microbial α diversity of mice in the DSS group decreased significantly relative to the CTRL group (*p* < 0.05). The diversity of colonic microorganisms in mice treated with CMP I and CMP II increased to some extent (*p* > 0.05), and the effect of CMP I was superior to CMP II to some extent (*p* > 0.05). The colon of the mammalian gastrointestinal tract has multiple sources of nutrition, making it a preferred site for many microorganisms [56]. Bacteroidetes are potential colonizers, and major players in maintaining the intestinal microbial food web, accounting for the majority of the intestinal flora [57,58]. They have been shown to reduce colitis in mice by altering the abundance and proportion of microorganisms in the colon. On the one hand, they increase probiotics in the colon, which break down polysaccharides to produce short-chain fatty acids (SCFA), which then bind to GPR43 on regulatory T cells (Tregs) [59], mediating a protective effect against colitis in mice. On the other hand, reducing the proportion of harmful bacteria in the colon reduced colitis in mice. Our results showed that the abundance of Proteobacteria and Parasutterella in the colons of the DSS-treated mice increased, while that of Bacteroidota, Muribaculaceae, and Verrucomicrobiota decreased. The abundance of Bacteroidota (*p* < 0.05), Muribaculaceae, and Verrucomicrobiota in the colon of the mice treated with CMP I increased, while the number of Proteobacteria and Parasutterella was diminished. The number of Bacteroidota, Muribaculaceae, and Lachnospiraceae increased in the colon of the mice treated with CMP II. CMP I and CMP II can regulate intestinal flora to treat intestinal inflammation in mice, and the therapeutic effect of CMP I is greater than that of CMP II, as evidenced by the expression quantity of various inflammatory factors and enzymes in the colon sections and sera of the mice, and by the proportional changes to the intestinal microorganisms in the mice. In general, the therapeutic effect of CMP I was superior to that of CMP II, both in terms of the expression levels of the various inflammatory factors and enzymes in the colon sections and sera of the mice and the proportion of intestinal microorganisms. On the one hand, CMP I may reduce colonic barrier permeability in mice by regulating the NF-kB metabolism and MAPKs signal transduction pathways. On the other hand, intestinal inflammation is improved by regulating the proportion and abundance of intestinal flora in the colon and by using probiotics to degrade them into short-chain fatty acids. Moreover, CMP I can also regulate the expressions of the MPO enzyme and SOD enzyme in mouse sera to prevent the peroxidation of colon tissue, thus reducing the symptoms of colitis in mice.

It has been shown that, on the one hand, the small molecular weight of polysaccharide is more easily degraded and utilized by microorganisms in the intestine; not only that, but it also shows greater immune-promoting function and antioxidant effects compared with high-molecular-weight polysaccharide [60,61,62]. On the other hand, the solubility of polysaccharide is also an important factor affecting its biological activity, and the solubility and biological activity of polysaccharide can be improved via chemical modification. The biological activity of modified polysaccharide is usually positively correlated with DS positive correlation [63,64]. There are two main differences between CMP I and CMP II: first, the molecular weights of CMP I and CMP II are 21 kDa and 68 kDa, respectively, and the molecular weight of CMP II is three times that of CMP I; second, for the same carboxymethylation modification method with different degrees of substitution, the degrees of substitution of CMP I and CMP II are 0.63 and 0.48, respectively. This indicates that the Poria polysaccharides produced through liquid fermentation are more easily modified than the natural nucleus Poria polysaccharide, which is more easily modified by carboxymethylation. Our results are consistent with the above findings; the molecular weight of CMP I was smaller than that of CMP II, while the degree of substitution was greater for CMP I than CMP II. Therefore, a possible reason for which CMP I has a greater therapeutic effect compared to CMP II is the difference in molecular weight and the degree of substitution. In this study, the repair effects of CMP I and CMP II on DSS-induced colitis in mice were investigated from the viewpoints of the degree of inflammation in the colon tissue, the expression levels of the inflammatory factors and enzymes in the sera, and the composition of the microbial community in the colon. In addition, the correlation between the abundance of microorganisms in the intestinal tract and the expression levels of various inflammatory factors and enzymes in the sera of mice was analyzed. It has been proved that both CMP I and CMP II can improve colitis in mice. However, there are shortcomings in this study: For the therapeutic effects of the two polysaccharides on colitis in mice, on the one hand, there is no explanation from the aspect of gene and protein expression for which genes guide the synthesis of proteins to achieve the therapeutic effect of colitis. Investigating which types of diversified molecular structures of the two polysaccharides exert their anti-inflammatory effects will be our next object of study.

## 4. Materials and Methods

The animal experiments were conducted in accordance with the regulations for the protection and utilization of laboratory animals at Hunan Agricultural University (Changsha, China), as authorized by the Biomedical Ethics Committee of Hunan Agricultural University (approval code: 2022114; approval date: 12 September 2022). A total of 30 8-week-old ICR female mice were obtained from Slaughter Jingda Company Laboratory Animal Central (Changsha, China). CMP I and CMP II were provided by the Institute of Agricultural Bioengineering, Hunan Agricultural University. The molecular weights were 21 kDa and 68 kDa, the degrees of substitution of CMP I and CMP II were 0.63 and 0.48, and the purity was higher than 85%. The impurities included water and soluble salt.

All the mice were placed in a standard environment (room temperature about 23 °C, 12 h cycle of alternating day and night). In order to mitigate the effects of environmental changes on the mice, the mice were first acclimatized and reared for 7 days. After the adaptation period, the mice were randomly assigned to five groups (Table 1, *n =* 6). During the experiment, all the mice were provided with adequate food and water. After 21 days, all the mice were made to fast for 12 h, and then their body weights were recorded. Next, after ether anesthesia and the collection of blood from the retro-orbital sinus, each mouse was killed through cervical dislocation. Finally, the colon was fixed with 4% formaldehyde, and the colon contents were collected and stored frozen with liquid nitrogen in a −80 °C refrigerator.

### 4.1. Colonic Histopathology

The colon tissue specimens were fixed with 4% formaldehyde, sliced, stained with hematoxylin and eosin, dehydrated with ethanol, and sealed with neutral gum. The histological morphology and tissue inflammation of the colons were observed and analyzed under a microscope (Motic, Beijing, China).

### 4.2. Detection of Inflammatory Cytokines in Colonic Tissue and Serum

The expression levels of inflammatory factors and two enzymes in colonic serum were detected using an ELISA kit according to the manufacturer’s instructions. Detailed ELISA kit information is shown in Table 2.

### 4.3. Microbial Community Analysis

The DNA of the samples was extracted using MN NucleoSpin 96 soil DNA kit (Shanghai Preferred Biotechnology Co., Ltd., Shanghai, China) in strict accordance with the manufacturer’s requirements. Primers F (5′-ACTCCTACGGGAGGCAGCA-3′) and R (5′-GGACTACHVGGGTWTCTAAT-3′) were used to amplify the high-variable region of bacterial 16S rRNA gene V3-V4. Then, Solexa PCR, nanodrop quantification, and mixed samples were performed. Finally, the column was purified and gelled for recovery. MiSeq high-throughput sequencing (Illumina, NovaSeq 6000, San Diego, CA, USA) was used.

### 4.4. Data Analysis

SPSS 21 was used to conduct the Levene test and Student test for the variance of the data, which satisfied the homogeneity test of variance. Origin 2021 Pro was used to draw the figures of body weight change, colon length, serum contents of various immune factors, MPO and SOD enzymes, and microbial content of different levels in the colon of the mice. The images were retouched using Adobe Illustrator 2021 (Adobe, San Jose, CA, USA). * *p* < 0.05 indicated significant differences.

## Figures and Tables

**Figure 1 ijms-24-09034-f001:**
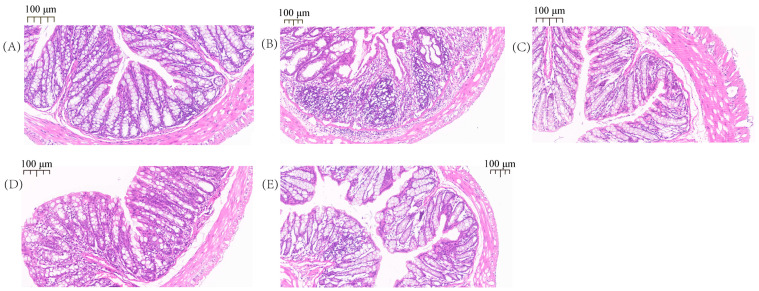
H&E staining results of mouse colon: (**A**) CTRL, (**B**) DSS, (**C**) SAZ, (**D**) CMP I, (**E**) CMP II.

**Figure 2 ijms-24-09034-f002:**
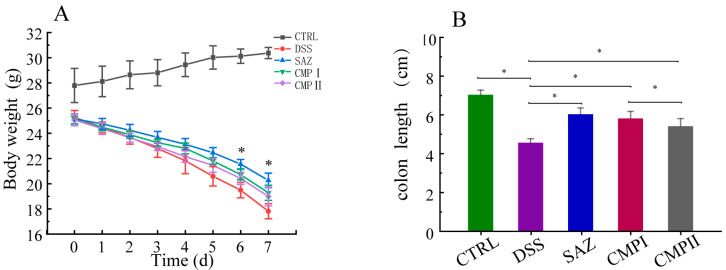
(**A**) Comparison of body weight changes in each group (*n* = 6), (**B**) final colon length of each group (*n* = 6). * indicates *p* < 0.05.

**Figure 3 ijms-24-09034-f003:**
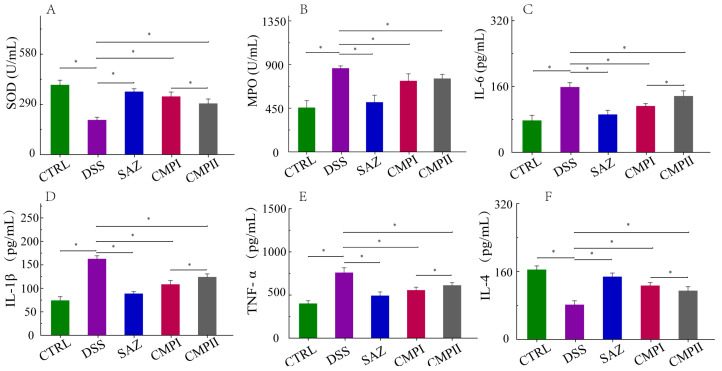
Expression of inflammatory factors and enzymes in sera of mice in each group (*n =* 6): (**A**) SOD, (**B**) MPO, (**C**) IL-6, (**D**) IL-1β, (**E**) TNF-α, and (**F**) IL-4. * indicates *p* < 0.05.

**Figure 4 ijms-24-09034-f004:**
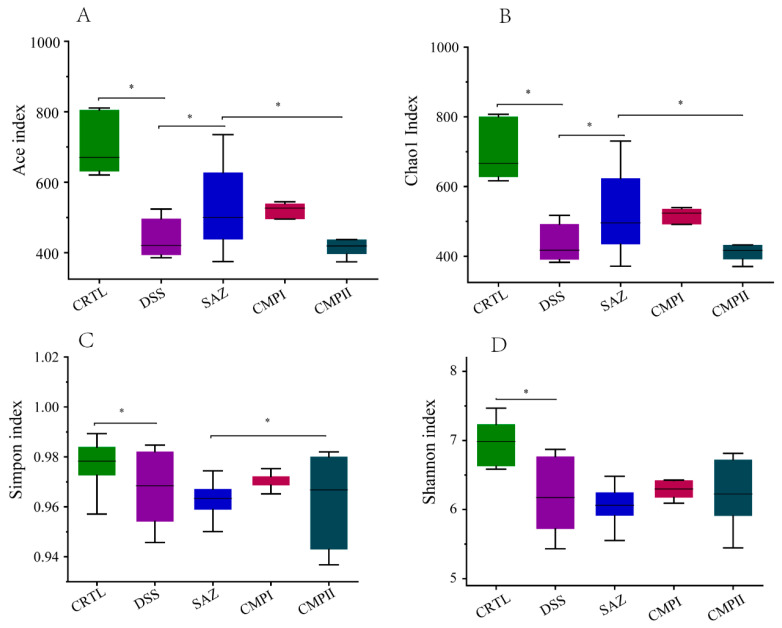
Colonic microbial α-diversity of mice in different groups (*n =* 6): (**A**) Ace index, (**B**) Chao1 index, (**C**) Simpson index, and (**D**) Shannon index. * indicates *p* < 0.05.

**Figure 5 ijms-24-09034-f005:**
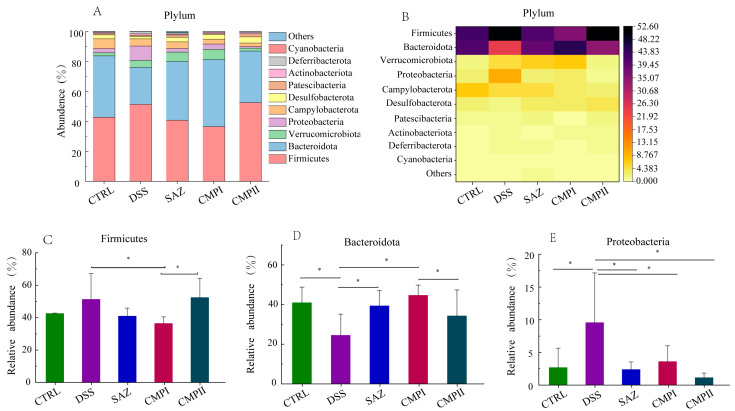
Effect of each group of treatments on microorganisms at the phylum level (*n =* 6): (**A**,**B**) top ten microorganisms with the percentage of phylum level, (**C**) Firmicutes, (**D**) Bacteroidota, (**E**) Proteobacteria. * indicates *p* < 0.05.

**Figure 6 ijms-24-09034-f006:**
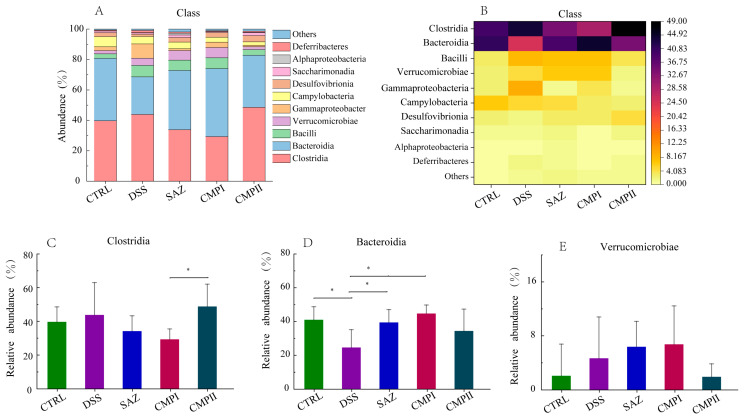
Effect of each group of treatments on microorganisms at the class level (*n =* 6): (**A**,**B**) top ten microorganisms in the class level, (**C**) Clostridia, (**D**) Bacteroidia, (**E**) Verrucomicrobiae. * indicating *p* < 0.05.

**Figure 7 ijms-24-09034-f007:**
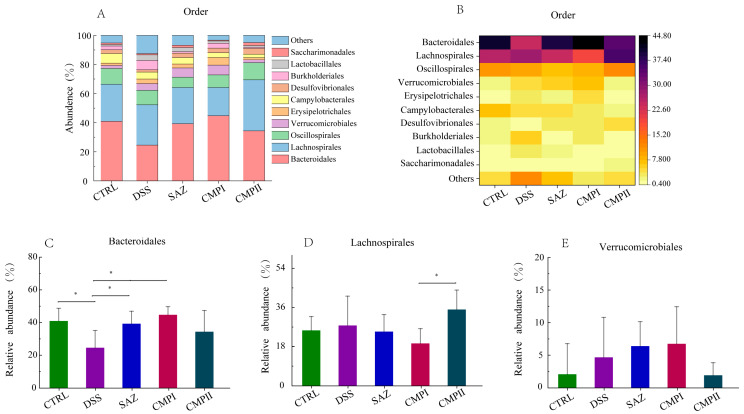
Effect of each group of treatments on microorganisms at the order level (*n =* 6): (**A**,**B**) top ten microorganisms with the percentage of order level, (**C**) Bacteroidales, (**D**) Lachnospirales, (**E**) Verrucomicrobiae. * indicates *p* < 0.05.

**Figure 8 ijms-24-09034-f008:**
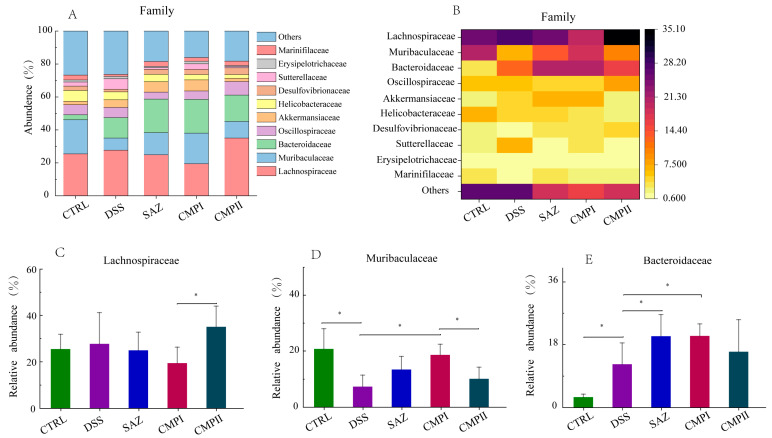
Effect of each group of treatments on microorganisms at the family level (*n =* 6): (**A**,**B**) top ten microorganisms in terms of family level content, (**C**) Lachnospiraceae, (**D**) Muribaculaceae, (**E**) Bacteroidaceae. * indicates *p* < 0.05.

**Figure 9 ijms-24-09034-f009:**
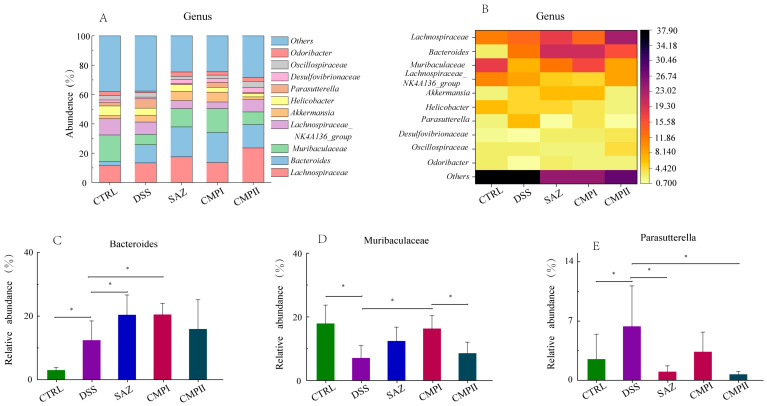
Effect of each group of treatments on microorganisms at genus level (*n =* 6): (**A**,**B**) top ten microorganisms with genus level content, (**C**) *Bacteroides*, (**D**) *Muribaculaceae*, and (**E**) *Parasutterella*. * indicates *p* < 0.05.

**Figure 10 ijms-24-09034-f010:**
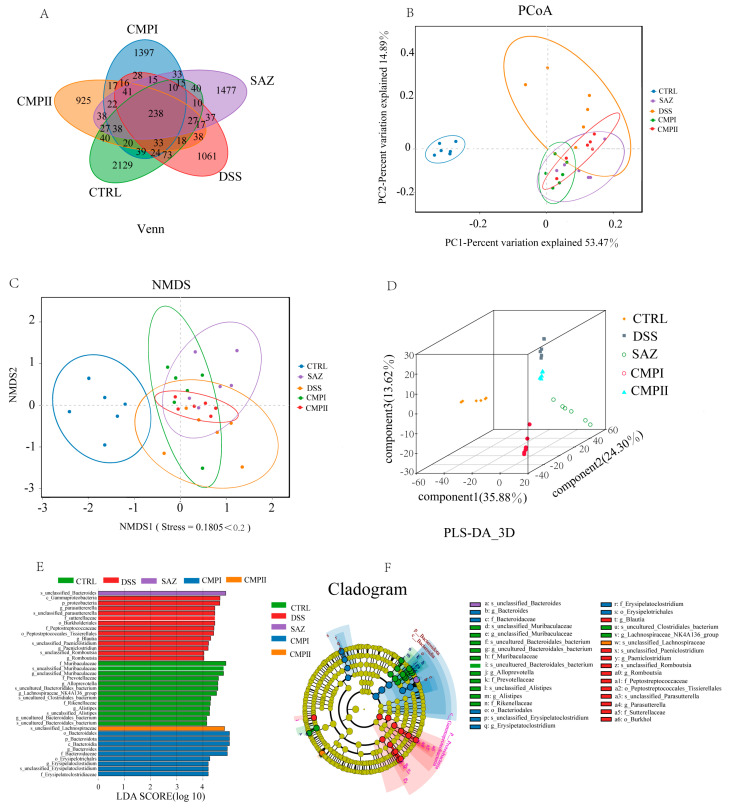
Beta-diversity and LEfSe analysis of the intestinal microbial flora: (**A**) Venn diagram, (**B**) PCoA, (**C**) NMDS, (**D**) PLS-DA 3D, (**E**,**F**) LEfSe.

**Figure 11 ijms-24-09034-f011:**
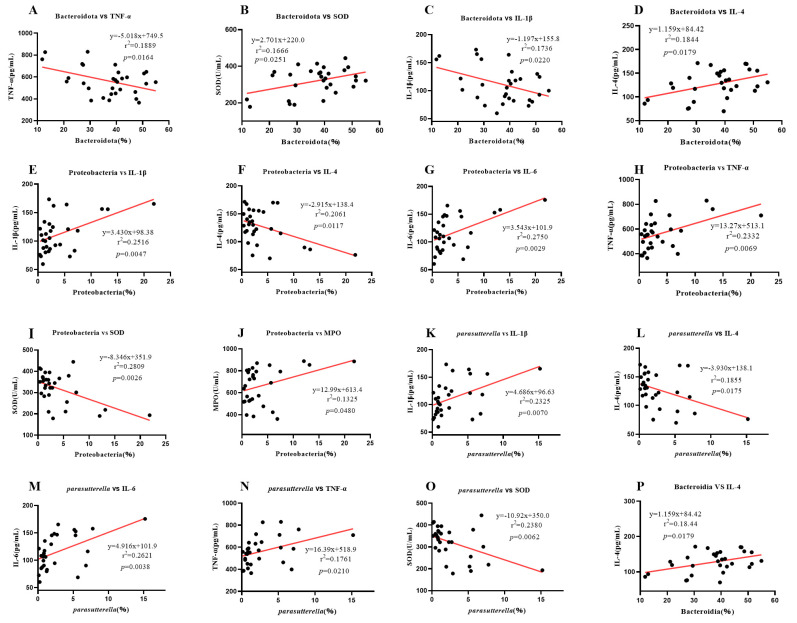
Correlation analysis between intestinal microbe difference and serum inflammatory factors and enzymes: (**A**) Bacteroidota vs. TNF-α, (**B**) Bacteroidota vs. SOD, (**C**) Bacteroidota vs. IL-1β, (**D**) Bacteroidota vs. IL-4, (**E**) Proteobacteria vs. IL-1β, (**F**) Proteobacteria vs. IL-4, (**G**) Proteobacteria vs. IL-6, (**H**) Proteobacteria vs. TNF-α, (**I**) Proteobacteria vs. SOD, (**J**) Proteobacteria vs. MPO, (**K**) *Parasutterella* vs. IL-1β, (**L**) *Parasutterella* vs. IL-4, (**M**) *Parasutterella* vs. IL-6, (**N**) *Parasutterella* vs. TNF-α, (**O**) *Parasutterella* vs. SOD, (**P**) Bacteroidia vs. IL-4.

**Table 1 ijms-24-09034-t001:** Group name and treatment.

Group Name and Abbreviation	Treatment
Control group: CTRL	Mice were gavaged with saline
Dextran sulfate sodium model group: DSS	Treatment with 5% DSS on days 8 to 14, followed by the instillation of sterile saline on days 15 to 21.
Sulfasalazine positive control group: SAZ	Treatment with 5% DSS on days 8 to 14, followed by the instillation of SAZ (300 mg/kg/day) on days 15 to 21.
Liquid fermentation carboxymethyl Poria polysaccharide: CMP I	Treatment with 5% DSS on days 8 to 14, followed by the instillation of CMP I (300 mg/kg/day) on days 15 to 21.
Natural sclerotium carboxymethyl Poria polysaccharide: CMP II	Treatment with 5% DSS on days 8 to 14, followed by the instillation of CMP II (300 mg/kg/day) on days 15 to 21.

**Table 2 ijms-24-09034-t002:** Detailed ELISA kit information.

Kit Name	Detection Range	Manufacturing or Marketing Company	Country
TNF-α detection kit	1.0–640 pg/mL	Shanghai Preferred Biotechnology	China
IL-1β detection kit	1–120 pg/mL	Shanghai Preferred Biotechnology	China
IL-6 detection kit	1–120 pg/mL	Shanghai Preferred Biotechnology	China
IL-4 detection kit	>1.0 pg/mL	Shanghai Preferred Biotechnology	China
SOD detection kit	>1.0 U/mL	Shanghai Preferred Biotechnology	China
MPO detection kit	>1.0 U/L	Shanghai Preferred Biotechnology	China

## Data Availability

The raw sequence data in this study are uploaded in the NCBI database, the accession is PRJNA936672 (https://www.ncbi.nlm.nih.gov/bioproject/PRJNA936672).

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
