# Peer review of "A Comparative Study on the Effects of Different Sources of Carboxymethyl Poria Polysaccharides on the Repair of DSS-Induced Colitis in Mice"

_ijms, 2023, doi:10.3390/ijms24109034_

Round 1

Reviewer 1 Report (Previous Reviewer 2)

The manuscript has been basically revised according to the comments. However, the authors should remark the changes in the resubmitted manuscript. In addition, the scientific English writing need to be significantly improved. For example:

1.     The space between Figure and number lacked and the figures should be arranged orderly.

2.     Abbreviations should be written in full when it was first mentioned. Please give the full spelling of DSS, SAZ, SOD, MPO etc. in the abstract.

3.     Figure 2A, the y-coordinate should be “body weight” instead of “body weight change”.

Author Response

See attached PDF with responses.

Reviewer 2 Report (New Reviewer)

Dear Editor,

I carefully read the revised version of the manuscript "Comparative study on the effect of different sources of carboxymethyl poria polysaccharides on the repair of DSS-induced colitis in mice".

My further comments and suggestions for the authors are the following:

 - English language needs to be carefully revised and improved. Currently, the manuscript is poorly written and this should be ammended during revision.

 - All the used abbreviations should be definted at their first occurrence in the manuscript (and in the abstract too).

 - Data analysis should be further and more deeply described. For example, the authors should specify how they assessed the normal distribution of the variables. Did they consider either 2-tailed or 1-tailed p-value< 0.05 as statistically significant?

 - How was the sample size calculated? This is a critical issue indeed.

 - In the discussion, the authors should further detail the limitations of their study.

Author Response

See attached PDF with responses to your concerns.

Reviewer 3 Report (New Reviewer)

The manuscript by Zhijie Tan et al. aimed to compare and study the therapeutic effect of different sources of carboxymethyl poria polysaccharides on DSS-induced ulcerative colitis. The study is of interest to the readers. However, the manuscript is poorly prepared and the study is too preliminary. The authors must critically revise the whole manuscript. I have the following questions and comments:

1, Extensive editing of the English language and style are required. The manuscript is poorly written. So many typos and errors. So many Chinglish. 

2, the chemical structure of the two carboxymethyl poria polysaccharides must be provided and relevant NMR and MS data must be added. 

3, the authors must further perform the PCA, NMDS, PLS-DA, PCoA, Veen, Lefse, Heatmap correlation  analyses to compare the structure of the gut microbiota and the key functional bacteria that were changed by polysaccharide treatment. The results and the study is too preliminary! 

IJMS is a good journal in this field and this manuscript is substandard to be considered for publication. I suggest to reject it. 

Author Response

See attached PDF with responses to your concerns.

Round 2

Reviewer 2 Report (New Reviewer)

Dear Editor,

I carefully read the revised version of the manuscript, that is improved in comparison with the original version. 

Reviewer 3 Report (New Reviewer)

The authors have revised the manuscript accordingly. It can be considered for publication. 

This manuscript is a resubmission of an earlier submission. The following is a list of the peer review reports and author responses from that submission.

Round 1

Reviewer 1 Report

Manuscript ID: ijms-2270221

Title: Comparative study on the effect of different sources of carboxymethyl poria polysaccharides on the repair of DSS-induced colitis in mice

In recent decades, many studies have shown that mushroom polysaccharides have anti-tumor, anti-inflammatory, and other biological activities.  The regulation of gut microbiota by mushroom polysaccharides has been also widely reported.

The paper titled, "Comparative study on the effect of different sources of carboxymethyl poria polysaccharides on the repair of DSS-induced colitis in mice", can be an important consideration for some investigators, but there are comments that need to be addressed by the authors.

(1) Please write the identification code and date of approval with the name of the ethics committee. Research procedures and animal welfare must be carried out in accordance with all relevant legislation.

(2) Please write how were the mice euthanized.

(3) Why did mice fast for 12 hours? (4 hours? 6 hours? 8 hours?)

(4) All information on all ELISA kits that were described in the manuscript is required -please write in brackets: abbreviation, cat. number, manufacturer, and range of detection. What is the range of detection in these ELISA kits?

(5) Please describe media and substances, which are used in the microbial analysis

(6). Please describe more information in the sections: "Colonic histopathology", "Detection of inflammatory cytokines in colonic tissue and serum", "Microbial Community Analysis" and " Data analysis"

Author Response

Dear reviewer

Please refer to the attachment

Reviewer 2 Report

In this work, Tan et al. reported using carboxymethyl poria polysaccharide to repair DSS-induced colitis in mice. The histological analysis, the levels of inflammatory cytokines, microorganisms in the colon etc. were assessed. The results indicated that therapeutic effect of CMPâ… on DSS-induced colitis in mice was superior to other groups. It can be accepted after addressing the below concerns. 

1. In Figure 1, there was no scale bar in the H&E staining imaging. And there just showed a representative picture, “n=6” in the caption made no sense. 

2. In Figure 2A, “baby weight” in the y-coordinate should be “body weight”. In addition, the average colonic length should be showed in the figures rather than one picture of one sample. 

3. It was a wrong caption in Figure 3. Please correct it. 

4. What is the meaning of DSS and SAZ? There was no explanation. And how much was the dosage of administration? Please specify the experimental detail. 

5. The mechanism of the therapeutic effect of CMPâ… should be studied or discussed in-depth.

6. What is the difference of CMPâ… extracted by liquid fermentation of Poria cocos and  CMPâ…¡ extracted by natural Poria cocos nuclei? Why is the therapeutic effect of CMPâ… better than that of CMPâ…¡? Please explain it. 

Author Response

(The authors gave the same response as above.)
